# Effects of human disturbances on wildlife behaviour and consequences for predator-prey overlap in Southeast Asia

Samuel Xin Tham Lee [1], Zachary Amir [1], Jonathan H. Moore [2,3], Kaitlyn M. Gaynor[4] & Matthew Scott Luskin [5,6]

Some animal species shift their activity towards increased nocturnality in disturbed habitats to avoid predominantly diurnal humans. This may alter diel overlap among species, a precondition to most predation and competition interactions that structure food webs. Here, using camera trap data from 10 tropical forest landscapes, we find that hyperdiverse Southeast Asian wildlife communities shift their peak activity from early mornings in intact habitats towards dawn and dusk in disturbed habitats (increased crepuscularity). Our results indicate that anthropogenic disturbances drive opposing behavioural adaptations based on rarity, size and feeding guild, with more nocturnality among the 59 rarer specialists' species, more diurnality for medium-sized generalists, and less diurnality for larger hunted species. Species turnover also played a role in underpinning community- and guild-level responses, with disturbances associated with markedly more detections of diurnal generalists and their medium-sized diurnal predators. However, overlap among predator-prey or competitor guilds does not vary with disturbance, suggesting that net species interactions may be conserved.

More than 75% of the Earth's land surface experiences measurable levels of anthropogenic disturbances[1], which impacts wildlife community composition and animal behaviour[2,3]. Quantifying the exact nature and magnitude of anthropogenic impacts on wildlife and how they affect species interactions like competition and predation remains a challenge[4,5]. One example is the impacts of humans on wildlife diel activity, or how species distribute their activity throughout the 24-h daily cycle. Wildlife may avoid humans due to historical and contemporary hunting that instilled a fear of humans as predators or aggressive competitors[6]. Over evolutionary time, species optimize their diel activity to balance trade-offs between obtaining resources (e.g., food, shelter), predation risks, and competition costs[7,8]. The predator-prey response race drives each to maximize fitness in the context of their environment and spatial and temporal niche of the

other[9,10]. Predators use sensory cues to seek and ambush prey, such as visual, auditory, and olfactory cues, and align their activity patterns to match their prey[11,12]. Meanwhile, competitors may repel each other through aggressive interactions leading to complementary activity patterns (i.e., temporal niche partitioning)[7,8]. Anthropogenic disturbances (e.g., forest edges, logging, light pollution, human presence, hunting) cause many animals to shift their activity patterns[13]. This shift alters temporal overlap among predators and competitors and likely alters species interactions (Fig. 1)[2,3].

Prior work suggests some terrestrial animals reduce activity during daylight to avoid diurnal humans[2,3] but it remains unclear whether activity shifts are consistent across species and if shifts are towards crepuscular periods (active at dusk and dawn) or nocturnal periods (no sunlight). This distinction is biologically meaningful, as crepuscular

[1]School of the Environment, University of Queensland, Brisbane, QLD, Australia. [2]School of Environmental Science and Engineering, Southern University of Science and Technology, Shenzhen, China. [3]School of Environmental Sciences, University of East Anglia, Norwich, United Kingdom. [4]Departments of Zoology and Botany, University of British Columbia, Vancouver, BC, Canada. [5]Institute of the Environment and Sustainability, University of California, Los Angeles, CA, USA. [6]Centre for Biodiversity and Conservation Science, University of Queensland, Brisbane, QLD, Australia. ✉e-mail: mluskin@ucla.edu

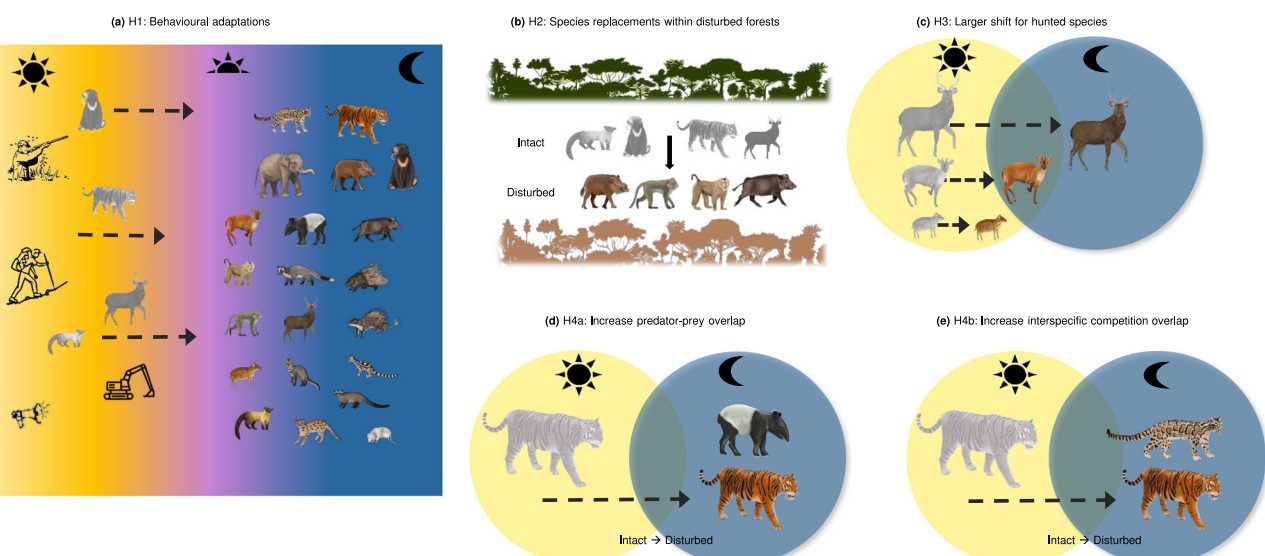

**Fig. 1 | Hypothesized wildlife behavioural response to humans and implications for species interactions.** Grey animals represent species that are present in intact forests but disappear or shift their activity in degraded forest. **a** Prior work suggests diurnal and crepuscular species alter their behaviour towards nocturnality in disturbed forests to avoid diurnal humans. The "sun", "half-sun" and "moon" symbols represent day, twilight, and night respectively. **b** Community- and guild-level temporal shifts may be driven by winner-loser species replacements within disturbed forests (e.g., loss of diurnal specialists or defaunation of game species alongside an increase in omnivorous generalists). The panel shows four habitat specialists in intact forests ("green" forest) being replaced by four generalists (pigs and macaques) within disturbed forests ("red" forest). **c** Larger hunted animals may show stronger avoidance of humans. **d**, **e** Increases in nocturnality may increase the temporal overlap of predator-prey and competitive species pairs. Dashed arrows denote the change in diel activity from day to night as forests become more disturbed while the solid arrow in panel **b** refers to the replacement of specialists by generalist species within disturbed forests.

and nocturnal periods offer different light and thermal environments[14]. Specifically, the semi-dark environment during crepuscular periods possesses moderate light and thermal conditions, potentially balancing the benefits of visibility for foraging and thermal stress associated with thermoregulation[14,15], but species can still be spotted by diurnal predators with vision adapted to light. Conversely, the darkness of nocturnal periods can provide a natural protection from light-adapted predators, however, species can incur fitness costs from thermoregulation and reduced foraging efficiency and difficulty spotting nocturnal predators due to lower visibility[14,15]. Therefore, behavioural adaptations to avoid diurnal humans can influence a species' ability to forage, evade predators, thermoregulate and ultimately survive.

Interspecific variation in animal behaviour is intimately linked to species traits and the likelihood of negative interactions with humans. Differences in guild-level responses have been reported, such as stronger avoidance of humans among large, hunted ungulates ('game species') than smaller animals or predators[16–19]. Here we compare activity patterns and temporal overlap for 10 wildlife communities in intact versus disturbed habitats and test the following hypotheses: (H1) species that are diurnal and crepuscular in intact habitats shift towards nocturnality in disturbed habitats to avoid diurnal humans, (H2) differences in the community- and guild-level activity pattern are driven by winner-loser species replacements[20] (e.g., intact habitat specialists that are diurnal being replaced by nocturnal generalists in disturbed habitats), (H3) hunters induce more fear in preferred species (larger game animals), driving stronger shifts compared to smaller and non-target species[21], and (H4) humans repel many guilds from diurnal hours, driving increased temporal overlap during nocturnal periods[22] among predator-prey and competitor species pairs (Fig. 1).

The potential for species interactions to occur depends, in part, on species being active at the same time and place, although co-occurrence (in space or time) does not necessitate interactions[23,24]. Humans, habitat loss, and habitat degradation increase the spatial overlap of species compared to intact areas[25], and are thus expected to increase the likelihood of interactions[25] due to reductions in area and environmental niche space. Similarly, higher temporal overlap is expected to be positively related to the likelihood of interactions[26]. Nonetheless, our interpretation of how temporal overlap may alter species interactions is limited to the 'potential likelihood of interactions'[26], which is useful for understanding community dynamics.

Southeast Asia is an ideal study system to evaluate human influences on wildlife behaviour, as it hosts one of the world's most biodiverse mammal assemblages and experiences high rates of human disturbance[21,27,28]. Over 70% of the region's forest are deforested in the last 50 years[29,30] and over 70% of the remaining forests lie within a kilometre of a forest edge[31]. Logging, edges, and fragmentation increase access for hunters[32], culminating in a bleak set of synergistic threats to the region's vertebrates[33]. The long-term conservation of robust Asian rainforest food webs depends on the perpetuation of species interactions within disturbed forests[34]. However, direct interactions are rarely observed, and dietary studies are limited due to dung quickly decomposing or being removed by dung beetles. It is therefore often necessary to infer potential interactions based on body-size ratios, feeding guilds, spatial co-occurrence and temporal activity overlap[24,35].

Here, we quantify how human influences may reshape Southeast Asian wildlife behaviour and temporal interactions. We use circular kernel density functions (hereafter 'activity distributions') to test for activity shifts between a binary comparison of intact and degraded forests, and multinomial logit mixed models to test changes in activity across the observed range of forest degradation. We also use the coefficient of activity pattern overlap from the kernel density estimates to test for changes in predator-prey and competitor species pairs between both types of forests. We differentiate intact versus disturbed forests using the Forest Landscape Integrity Index (FLII or 'forest integrity' hereafter), which is the most comprehensive index available that captures observed pressures (e.g., human densities, infrastructure, agricultural landscapes, forest cover loss), and inferred

human pressures (e.g., forest edges and fragments, and alterations in forest connectivity)[21]. We show that the community-level shift towards reduced nocturnality in more disturbed areas is mediated predominantly by changes in species composition (more activity from diurnal generalist omnivores and less activity from specialists) with a smaller effect of species adapting their behaviour. We also found that guild-level responses are highly variable, with large animals (>40 kg) and predators showing stronger responses than small-to-medium-sized animals.

## Results

We assess activity patterns using 31,138 independent detections from 20 camera deployment sessions at 10 forested landscapes in Thailand (two landscapes), Peninsular Malaysia (two landscapes), Sumatra (three landscapes), Singapore (one landscape), and Malaysian Borneo (two landscapes) using 1218 cameras (58,608 trap nights; Fig. 2; Supplementary Table 1). We include all vertebrates >1 kg in the guild- and community-level analyses totalling 57 mammalian species (excluding humans), four terrestrial bird species and two reptiles (Supplementary Table 2). For the species-level analyses, we only include species with ≥15 detections within each diel category (i.e., day, twilight and night) totalling 14 mammals and one terrestrial bird (Supplementary Table 3). In the species pair overlap analyses, we require detections of both species to be ≥20 detections in both intact and disturbed forests, which result in 21 mammals and two terrestrial bird species with sufficient sample sizes (29,879 detections retained; Supplementary Table 4). Of the 23 species meeting these criteria, there are six carnivores, 10 herbivores, and seven omnivores. We further group these guilds into different size classes where we define large animals as >20 kg (12 species), medium animals as 4–20 kg (26 species), and small animals as <4 kg (25 species). Just four medium and large omnivore

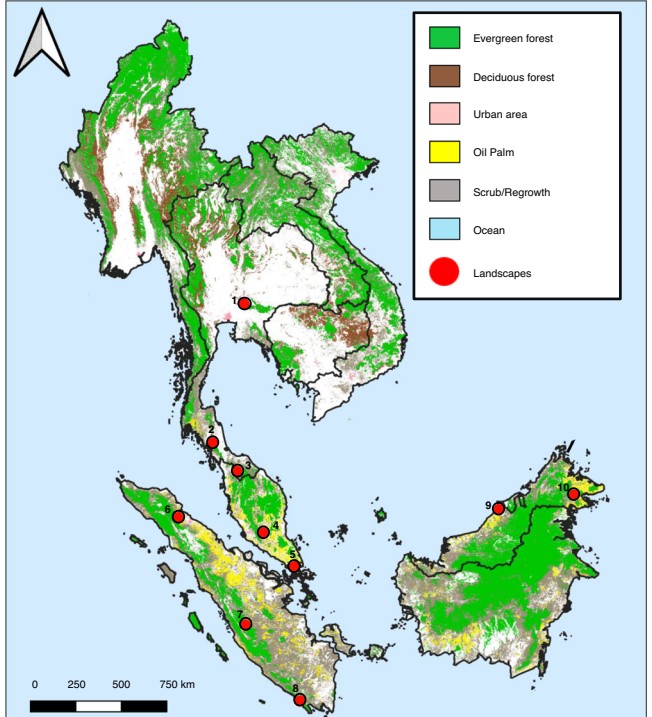

**Fig. 2 | Landscape-scale camera trapping was undertaken from 2013 to 2020 at ten Southeast Asian rainforests.** Different colours indicate different land uses and red dots are the locations of landscapes where camera trapping surveys are conducted (i.e., 1 = Khao Yai National Park, 2 = Khao Chong Nature Reserve, 3 = Ulu Muda Forest Reserve, 4 = Pasoh Forest Reserve, 5 = Singapore, 6 = Mount Leuser National Park, 7 = Kerinci Seblat National Park, 8 = Bukit Barisan Selatan National Park, 9 = Lambir Hills National Park and 10 = Danum Valley Conservation Area).

species – bearded pigs (*Sus barbatus*), wild boar (*S. scrofa*), long-tailed macaques (*Macaca fascicularis*) and pig-tailed macaques (*M. nemestrina*; hereafter just 'pigs and macaques') – account for 39% of all detections in intact forests but 68% in the most disturbed forests (Fig. 3). To account for the outsized effect of this subset of abundant pigs and macaques in dictating overall trends, we present community-level analyses including and excluding these four species, the latter group containing 14,635 detections of the other 59 species, which we will call 'rarer specialists' hereafter for simplicity, noting that there is a wide range of species types and traits included within this group.

### Community- and guild-level behavioural shifts using kernel density estimation

There is a significant shift in community-level activity when moving from intact to disturbed forests, which induces slightly more diurnal (+1.2%) and crepuscular (+2.0%) detections and a decrease (−3.2%) in nocturnal detections ($p < 0.001$; Fig. 3a). In the disturbed forests, the overall activity peak (AP) shifts an hour earlier in the morning (from $AP_{intact} = 0758$ h to $AP_{disturbed} = 0701$ h towards dawn and thus more crepuscular). The secondary activity peak at dusk is also more pronounced in disturbed forests. The community-level shift to diurnal and crepuscular hours is driven by the two most common pig and macaque species and operates through two mechanisms: first, pigs and macaques shift their behaviour in disturbed forests towards diurnality (+6.4% more diurnal detections; $p < 0.001$; Fig. 3c, Supplementary Figs. S5 and S6), and second, there are 76% more detections of these primarily diurnal and crepuscular pigs and macaques in disturbed than intact forests (i.e., species replacements explained overall trends). When pigs and macaques are excluded from the analyses, the activity pattern for the other 59 species are generally crepuscular within both forest types ($AP_{intact} = 0633$ h; $AP_{disturbed} = 1814$ h) but with +6.9% detections during nocturnal hours in disturbed habitats ($p < 0.001$; Fig. 3b).

In our guild-level kernel density analyses, large carnivores (>20 kg) show the strongest changes in disturbed forests by reducing early morning activity (when humans become active) and becoming more cathemeral (i.e., no distinct activity peak). This is especially true of tigers, with more detections during nocturnal (+16.5%) and crepuscular hours (+5.4%) in disturbed forests ($p = 0.016$; Fig. 3d; Supplementary Tables 5 and 6). Medium carnivores (4–20 kg), on the other hand, shift their peak activity towards diurnality (+25.8% detections; $p < 0.001$; Fig. 3g) while small carnivores largely remain nocturnal in both forest types, ($p = 0.11$; Fig. 3j). Large and small herbivores remain predominantly crepuscular in both forest types ($p > 0.05$; Fig. 3e, k) while medium herbivores become less crepuscular and more nocturnal in disturbed forests (+13.1%; $p < 0.001$; Fig. 3h). Large and medium omnivores – which includes both pig and macaques – increase their crepuscular activity within disturbed forests at the guild and species level ($p < 0.001$ for all comparisons; Figs. 3f, i and 4). Small omnivores increase diurnal (+6.1%) and crepuscular (+3.9%) detections but this is not statistically significant ($p = 0.25$; Fig. 3l).

Kernel density significance testing assesses the overlap of the two distributions, as opposed to identifying if there are significant changes within a diel category (e.g., significantly more diurnal, while non-significant changes in nocturnality and crepuscularity) and lack statistical accommodations for species-level random effects or nested sampling designs. We therefore also use multinomial logit mixed model analyses (MNLMMs) with species- and landscape-level random effects.

We also assess the sensitivity of our results to splitting cameras into low versus high disturbance using the median forest integrity value. Next, we re-analyse activity patterns by comparing 1st and 3rd quartiles of forest integrity (i.e., very intact versus very disturbed). This more extreme split exhibits qualitatively similar results but more pronounced changes, such as medium carnivores increasing their diurnal detections by +51% within the most disturbed forests

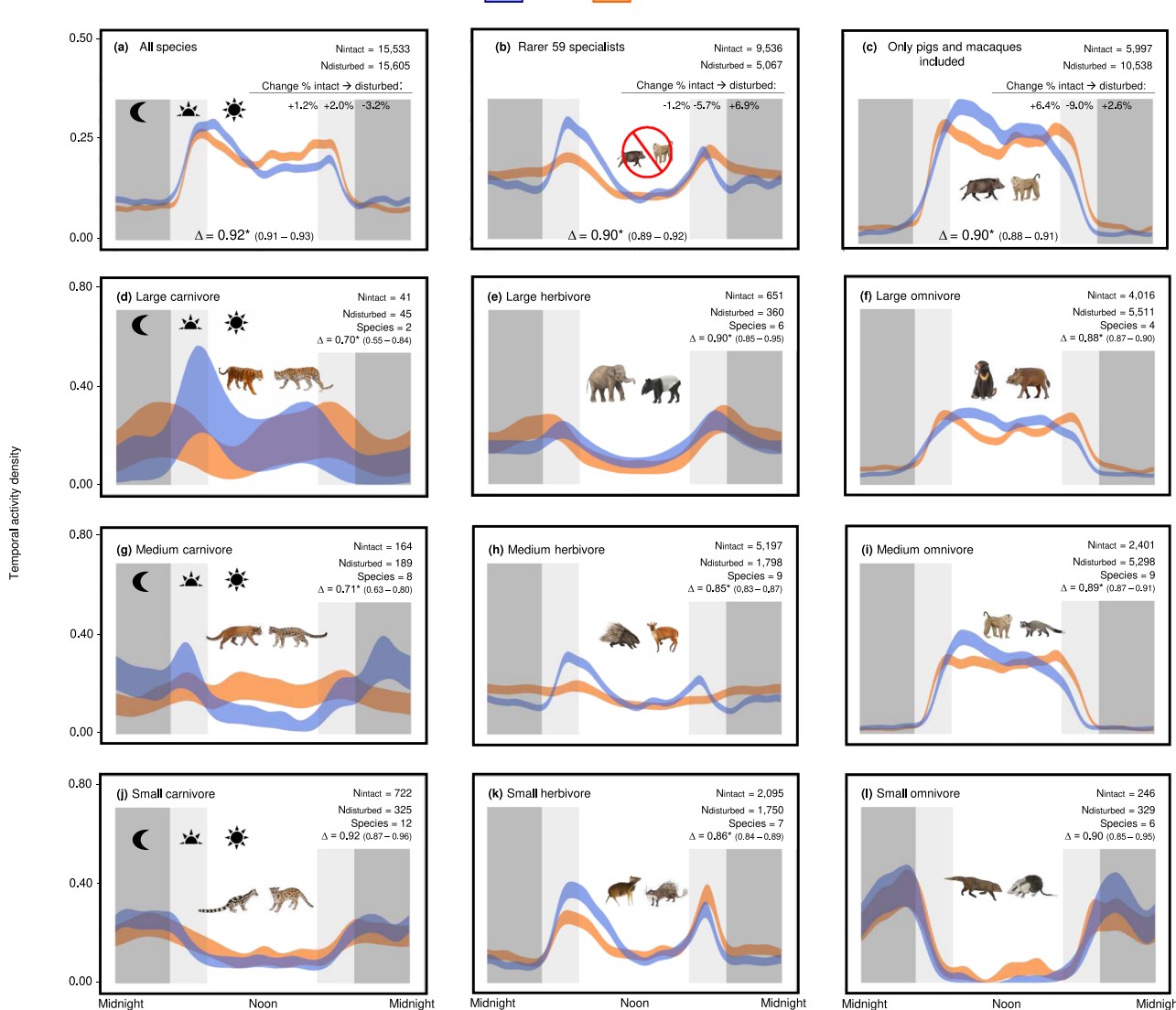

**Fig. 3 | Wildlife activity in intact and disturbed forests of Southeast Asia.** The blue ribbon shows detections from cameras in intact forest (FLII > 8.77) and the orange shows detection from cameras located in comparatively disturbed forests (FLII ≤ 8.77). $N_{intact}$ and $N_{disturbed}$ show sample sizes within intact and disturbed forests respectively, the Δ 'delta' denotes the overlap among the two activity distributions, and the * asterisk denotes the *p*-value estimated using the two-sided compareCkern test (Supplementary Table 6), however, we do note that no further statistical adjustments were made for each of the comparisons. "Change % intact → disturbed" is calculated within each diel category, i.e., (diurnal detections in disturbed habitats/total detections in disturbed habitats · diurnal detections in intact habitats/total detections in intact habitats) *100. Background shading denotes diurnal hours (white), crepuscular hours (light grey) and nocturnal hours (dark grey). **a** Community-level temporal activity pattern for all 63 vertebrate species. The increase in crepuscular activity in disturbed habitats is not driven by species adapting their behaviours but species replacements, namely a pronounced increase in the pigs and macaques that are active in diurnal and crepuscular periods in both intact and disturbed forests. **b** Community-level temporal activity pattern when the four most abundant pigs and macaque species are excluded. **c** Diel activity of the four most dominant species that account for 68% of detections in disturbed forests. **d–l** Guild-level wildlife activity patterns in intact and disturbed forests. Ribbons show 95% confidence intervals from bootstrapped activity distributions. The exact *P*-values for each comparison are as follows (Note that the compareCkern function outputs "0" when *P* < 0.001, which we report here as "*P* < 0.001"): **a** *P* < 0.001, **b** *P* < 0.001, **c** *P* < 0.001, **d** *P* = 0.016, **e** *P* = 0.023, **f** *P* < 0.001, **g** *P* < 0.001, **h** *P* < 0.001, **i** *P* < 0.001, **j** *P* = 0.11, **k** *P* < 0.001, and **l** *P* = 0.25.

compared to +25.8% when splitting by median forest disturbance (Supplementary Fig. 4; Supplementary Table 5).

## Community and guild-level behavioural shifts using multinomial logistic regressions

We fit multinomial logit mixed models (MNLMMs) to assess how the likelihood of wildlife detections occurring in diurnal, twilight, or nocturnal periods varied with disturbance and species traits. We include a species-level random effect so that the model reflects behavioural changes within species and a landscape random effect to account for our nested sampling design. When all species are considered together, we find that the top model includes a three-way interaction between disturbance, body size, and feeding guild, suggesting the effects of disturbance vary with body size and feeding guild (Supplementary Table 7). For all species, there is also a significant net effect of disturbance increasing nocturnality (Fig. 5a; Supplementary Table 12). We repeat the MNLMMs for the 59 rarer specialists and the four generalist pig and macaque species and neither show significant net effects of disturbance mediating diel activity (Fig. 5b, c; Supplementary Tables 13 and 14). However, focusing on this level of statistical significance overlooks biologically important differences among the specialists versus generalist groupings.

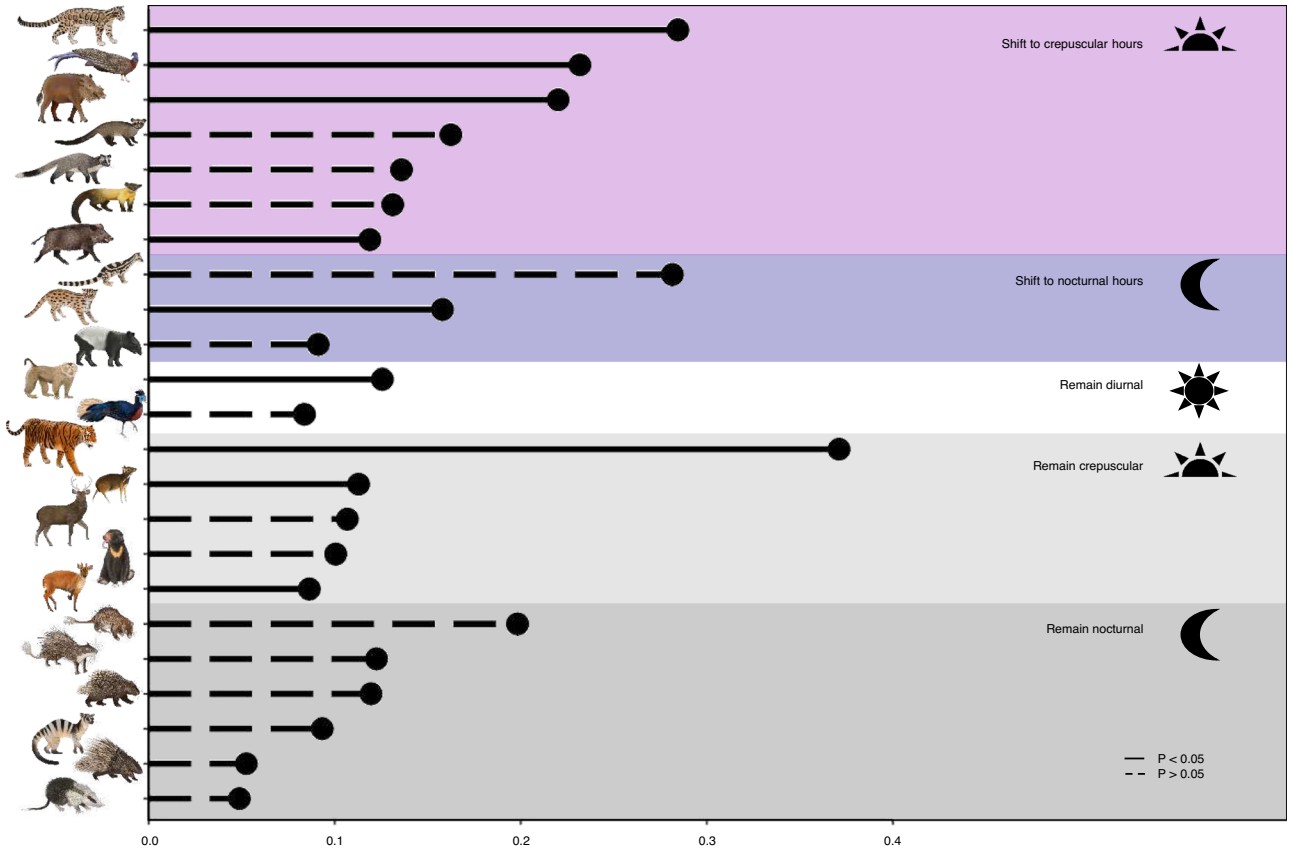

**Fig. 4 | Lollipop diagram summarising the species-level temporal shifts between intact (FLII > 8.77) and disturbed (FLII ≤ 8.77) forests from the kernel density analyses.** Solid lines show significant shifts while dashed lines indicate non-significant shifts. The colour bands refer to whether there was shift in peak activity between intact and disturbed forests. The species shown from top to bottom are clouded leopard (*Neofelis spp.*), great argus (*Argusianus argus*), bearded pig (*Sus barbatus*), common palm civet (*Paradoxurus hermaphroditus*), masked palm civet (*Paguma larvata*), yellow-throated marten (*Martes flavigula*), wild boar (*Sus scrofa*), banded linsang (*Prionodon linsang*), leopard cat (*Prionailurus spp.*), Malay tapir

(*Tapirus indicus*), southern pig-tailed macaque (*Macaca nemestrina*), crested fire-back pheasant (*Lophura spp.*), tiger (*Panthera tigris*), mouse deer (*Tragulus spp.*), sambar deer (*Rusa unicolor*), sun bear (*Helarctos malayanus*), red muntjac (*Muntiacus muntjak*), long-tailed porcupine (*Trichys fasciculata*), Asiatic brush-tailed porcupine (*Atherurus macrourus*), Bornean porcupine (*Hystrix crassispinis*), banded civet (*Hemigalus derbyanus*), Malayan porcupine (*Hystrix brachyurus*) and moon rat (*Echinosorex gymnura*). For exact *P*-values and difference in overlap coefficients (1-Δ), please refer to Supplementary Table 19.

Namely, the rarer 59 specialists' probability of nocturnality in the most disturbed sites [$Pr$(night) at FLII of 0] is 4.6 times higher than for pigs and macaques [$Pr$(night)$_{rarer\_specialists}$ = 0.41, CI = 0.39–0.43; $Pr$(night)$_{pigs\_macaques}$ = 0.089, CI = 0.066–0.11], and specialists' nocturnality increase in the most intact forest (FLII of 10) to 6.4 times higher than pigs and macaques [$Pr$(night)$_{rarer\_specialists}$ = 0.46, CI = 0.44–0.48; $Pr$(night)$_{pigs\_macaques}$ = 0.072, CI = 0.052–0.092].

There is also high variation among guilds, with an opposing size-mediated influence of disturbance on omnivores and carnivores. Specifically, disturbances drive significant shifts away from diurnality for larger carnivores and omnivores that are often hunted, while medium-sized carnivores and omnivores that are less hunted become more diurnal (i.e., macaques; Fig. 5d, f, g, i; Supplementary Table 17).

We also test the extent to which results are driven by species replacements by repeating the MNLMM analyses after removing species-level random effects, thereby treating all detections within guilds equally regardless of species (Supplementary Fig. 2; Supplementary Table 16). When ignoring species identity, disturbance no longer increase overall nocturnality (all species) and this due to the pronounced increase in probability of diurnal detections in medium-sized omnivore detections (Supplementary Fig. 2i; Supplementary Table 12). Disturbance also induces significant increases in the diurnality of medium-sized carnivores (Supplementary Fig. 2g), suggesting

species replacements are driving these guild-level trends (i.e., diurnal golden cat and yellow-throated marten replacing nocturnal clouded leopards).

## No changes in potential interactions
We calculate temporal overlap (Δ) for species pairs in both intact and disturbed forests to test for changes in the potential for interactions that may be more likely for species active at the same time. Rules for defining predator-prey and competitor pairs are based on guild and body size (Section 2.7 of Methods; Supplementary Tables 20 and 22). There is no evidence that overlap significantly differs between intact and disturbed forests when considering all potential competitor species pairs ($t_{68}$ = 0.22, $p$ = 0.83; Fig. 6a, b) or all potential predator-prey species pairs ($t_{138}$ = 0.85, $p$ = 0.40; Fig. 6c, d). There are no significant shifts when considering predator-prey or competitor sizes (Supplementary Tables 21 and 23).

## Discussion
Wildlife behaviour, including diel activity patterns, is an important determinant of species interactions and niche partitioning within ecological communities. Our findings from Southeast Asian forests suggest that human disturbances induce directional changes in wildlife communities via both behavioural shifts and species turnover.

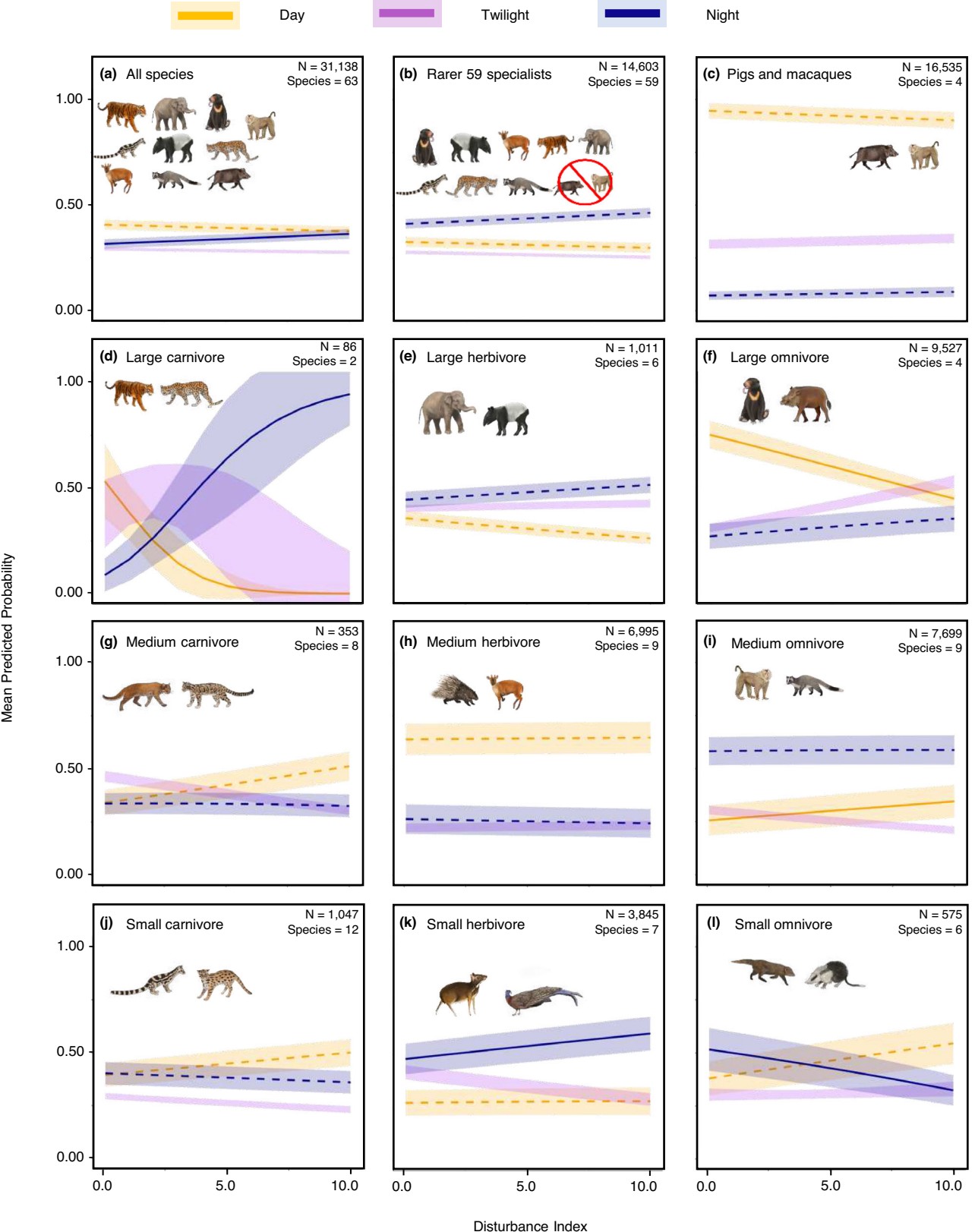

Specifically, the 59 rarer specialist species yield fewer detections and become more nocturnal in disturbed areas, providing support for H1 (Fig. 1). There is a reduction in diurnal activity for larger animals that are often most targeted by hunters, while small-sized animals that are infrequently targeted by hunters show weaker responses, providing support for H1 and H3. There is also a pronounced rise in detections of four common generalist species (pigs and macaques) from 39% of all detections in intact habitats to 68% in disturbed areas, providing strong support for H2 on species turnover. The net effect is that wild animal activity shifts from daytime peaks in intact forests towards dawn and dusk activity in disturbed forests. However, although species-specific pairwise temporal overlap changes in disturbed areas,

**Fig. 5 | Influence of humans and forest disturbance on diel activity.** Multinomial logistic mixed-effect models include forest integrity as a fixed effect and species and landscape as random effects with trend lines corresponding to the mean predicted probabilities of diel activity occurring during the day, twilight and night (orange, purple and blue lines, respectively). **a** The likelihood of detections for all 63 species. Results for the rarer 59 specialists (**b**) − whose combined detections decreased in disturbed areas − are shown separately from the four most abundant pig and macaque species (**c**) − whose detections increased in disturbed areas.

**d**–**l** Guild -specific diel activity responses. The effect of humans and habitat degradation is estimated using the inverse of the forest integrity (FLII)[21], which we call the 'disturbance index'. N is the independent detections of each community or guild. Solid lines show trends for day and night detections that have statistically significant slopes than twilight detections, the reference category. Dashed lines show non-significantly different trends and shaded areas show 95% confidence intervals.

Competitor-competitor temporal overlap

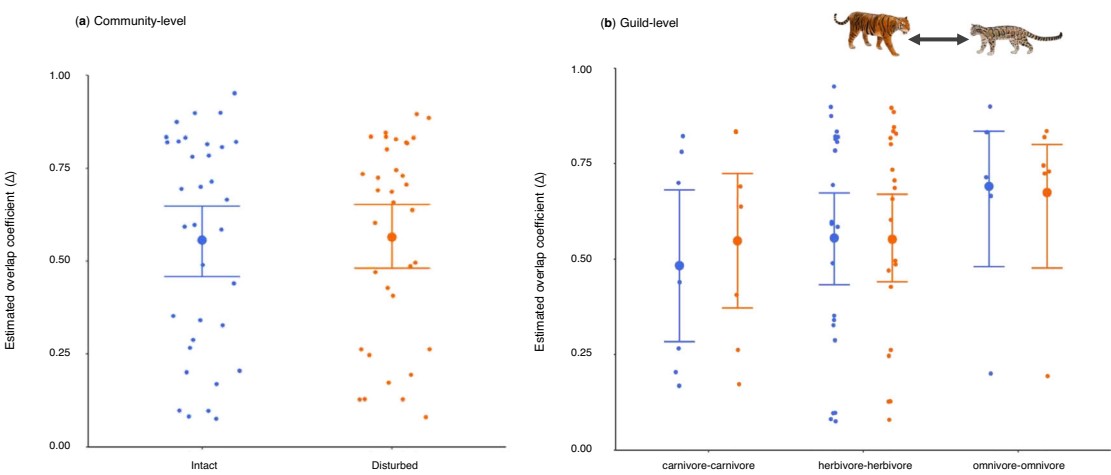

Predator-prey temporal overlap

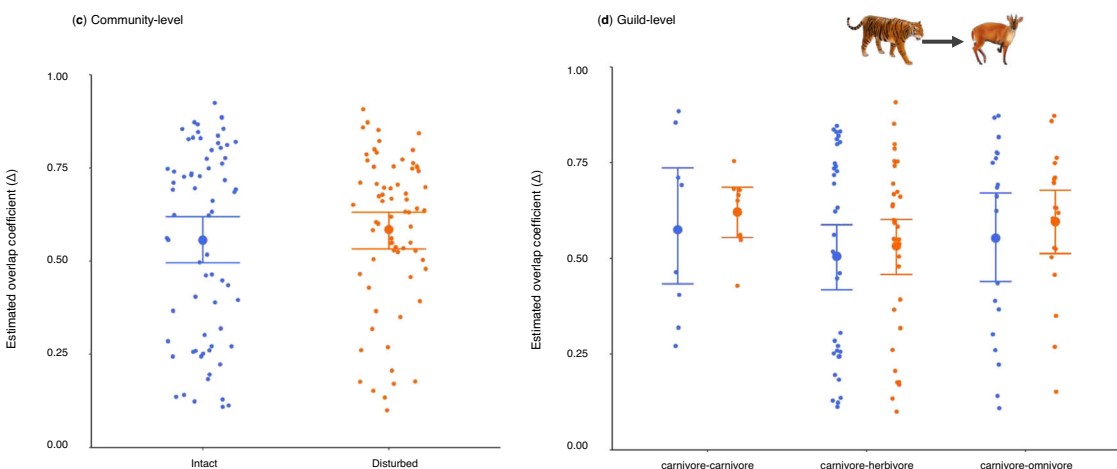

**Fig. 6 | No significant changes in potential competitive or predatory interactions between intact and disturbed forests.** The y-axis shows overlap (Δ) in activity distributions among competitor pairs, summarized at the community-level (**a**) and guild-level (**b**) in intact (blue) and disturbed (orange) forests. The same interpretation applies for the predator and prey species in panels **c** and **d**. The error bars show the 95% confidence interval and the larger dots show the mean Δ for each grouping. All community-level differences are evaluated for significance using a paired t-test comparing species pairwise overlap values in intact and disturbed forests ($n_{predation} = 138$; $n_{competition} = 68$). All guild-level differences are evaluated by

a Mann−Whittney U-test comparing pairwise overlap values for species of the same guild for competition (**a**, **b**) or predator-prey pairs (**c**, **d**) in intact and disturbed forests ($n_{predation} = 138$ pairs; $n_{competition} = 68$ pairs). We also note that all the statistical tests are two-sided and no further statistical adjustments were conducted for each community- and guild-level pairwise comparisons. Only the 23 species with >20 detections in both intact and disturbed forests are included in species overlap analyses. The full interaction matrix for species pairs undergoing competition and predation is provided in Supplementary Tables 20 and 22, respectively.

there are no significant shifts in the overall temporal overlap of predator-prey and competitors. This outcome does not support H4 that predicts humans and disturbances are driving altered species interactions at the community scale, and we note these results are consistent with prior work[22,36].

The effects of anthropogenic disturbances on wildlife behaviour vary across taxa[2,13,19,37] and our study highlights the nuanced diel

activity responses among guilds. Namely, disturbance increases the likelihood of nocturnal detections in rarer specialists' species and the likelihood of diurnal detections in medium-sized but not large generalist omnivores (behavioural adaptations) and an increase in the absolute number of detections of generalists and their medium-sized predators (species turnover). The large differences in responses of specialists versus generalists aligns well with recent work on tropical

forest mammal occupancy and abundance[20,38,39]. Differences in the magnitude of guild- and species-level shifts may arise, in part, from differential hunting pressure across taxa and locations. For example, bearded pigs are commonly hunted in their core range of Borneo and show strong shifts away from diurnal activity in disturbed habitats (Fig. 4; Supplementary Figs. 5 and 6), while the ecologically similar wild boars − which are absent from Borneo and rarely hunted in Peninsular Malaysia and Sumatra due to Halal diet taboos − show smaller shifts[40,41]. This difference indicates the mechanism driving some behavioural changes in pigs is likely the fear of physical harm from hunting, as opposed to effects from altered habitat conditions, noises, or smells. For most species, however, our kernel density and MNLMMs results show that the magnitude of behavioural change moving from intact to disturbed areas is relatively constrained. Specifically, the kernel density analysis finds only three species of 23 altered their peak activity from diurnal to crepuscular (bearded pigs, wild boars, and great argus pheasants) and one species − the leopard cat − significantly shifts from crepuscular to nocturnal.

The effects of disturbance on predator behaviour may be mediated by the responses of their prey, or vice versa. For example, clouded leopards shift from nocturnal in intact forests towards peak activity near dawn within disturbed forests (0600 h; Supplementary Table 19), likely adapting to an altered prey base, and indeed their overlap with some pigs, macaques, and the great argus pheasants increase in disturbed areas (Supplementary Table 22)[42–44]. Likewise, leopard cats' shift from crepuscular to nocturnality has previously been reported to hunt nocturnal rodent crop pests in disturbed forest edges[45–48]. The increase in diurnal activity of medium-sized carnivores may be tracking more medium-sized omnivores in disturbed areas, or these animals may be responding to competitive release and niche partitioning to avoid larger competitors, which have a reduction in diurnality in disturbed areas. Linking these behavioural adaptations and changes in predator-prey overlap to actual predation patterns requires further research on spatial overlap and predator diets in intact and degraded forests.

We report smaller effects of human disturbance on diel activity than some studies. This difference may be partly due to contemporary Asian vertebrate communities having undergone prior filtering that has removed the most sensitive species[20,34,49,50]. Historic extinction filters may have produced contemporary communities with comparatively disturbance-tolerant animals since they survived fluctuating habitat conditions and contiguous areas (i.e., due to sea-level changes) and have coexisted with human hunters for almost 45,000 years[34,49,50]. This idea is supported by recent studies that documented low spatial avoidance of humans or disturbed habitats by megafauna in Southeast Asia, and a high proportion of habitat generalists in the Asian rainforests compared to the other major tropical regions[34,49]. Our conservative results may also be due to a balance between animals avoiding human activity and costs to fitness for being active during suboptimal light conditions[51,52]. Many mammal species have numerous adaptations to light (corneal size and opsin proteins[51,52]) so the intermediate luminosity of twilight is preferable to night. Our results support other recent work finding that humans induce relatively small changes in wildlife behaviour at the species level (e.g., shifts from diurnal to cathemeral or crepuscular, as opposed to shifts to diurnal or nocturnal)[17,18]. Further work in other systems is needed to better understand how both recent and evolutionary history shape differences in the magnitude of change in diel activity in response to human disturbance.

These findings have the potential to inform conservation planning and mitigate undesirable effects of human disturbance on wild animals. Options to limit the negative impacts of humans on wildlife may include noise or luminosity zoning, analogous to spatial zoning, in and around protected areas that restrict human activities when species of high conservation priority are active[2]. In particular, protected areas can establish 'open' and 'closed' seasons to coincide with important wildlife events, cues or timings, such as breeding seasons, hibernation periods and migration of endangered species[2]. Our results also suggest the disturbed forest wildlife communities in Southeast Asia are dominated by diurnal and crepuscular medium-large omnivores, namely pigs and macaques, which is consistent with site-specific work[53,54]. The abundance of these generalist species can potentially pose severe risks to farming, livestock, and human health[27,53,55] and management may need to be considered. Finally, there are exciting and important opportunities to assess how the loss of a particular species can affect interacting species behaviour, such as how carnivore diel activity responds to the influence of African Swine Fever that is removing Asia's wild pigs, or a potential decline in macaques due to poaching for the medical trade[56,57].

## Methods

### Study area and sampling design
We sample wildlife with camera traps at 10 landscapes in Thailand, Peninsular Malaysia, Singapore, Sumatra, and Malaysian Borneo. These landscapes are dominated by tropical evergreen lowland or hill rainforests with canopy emergent trees dominated by the Dipterocarpaceae family (Fig. 2; see Supplementary Table 1 for specific landscape characteristics). We deploy 18–78 passive infrared Bushnell and Reconyx cameras for 60–90 days covering 10–813 km² at each landscape. We standardise deployment methods across all landscapes by spacing cameras >500 m in larger forests (>50 km²) and 100–500 m apart in smaller forest patches (e.g., Singapore) and attaching them to trees 0.3 m above ground along natural wildlife trails or hiking paths. Independent detections of the same species occurr when images are more than 30 min apart. We correct for time zones, sun angle in the sky and seasonal variation by standardising sunrise ($\pi/2$ or 1.571 radians) and sunset ($3\pi/2$ or 4.712 radians) using the suntime() function in the 'overlap' package version 0.3.4[58,59]. We then define and summarize each independent detection into day, twilight and night categories. Day includes detections occurring between 0730 and 1630 h (9 h total), twilight includes detections occurring between 0430–0730 h or 1630–1930 h (6 h total), and night includes detections occurring between 1930 and 0430 h (9 h total).

### Forest Landscape Integrity Index as a disturbance proxy
We use the Forest Landscape Integrity Index (FLII or 'forest integrity' hereafter) as a proxy for the many different direct and indirect impacts that humans may have on wildlife. This is the most comprehensive index available that captures a variety of direct and indirect human influences in a geographically consistent manner and is standardized globally, which is important for future work replicating our approach[21]. Forest integrity incorporates anthropogenic pressures from observed human pressures like human densities, infrastructure, agricultural landscapes, and forest cover loss, as well as inferred human pressures from forest edges, fragmentation, and connectivity[21]. These components are then weighted and a final measure is standardized between 0 and 10, ranging to the most disturbed (highest human pressure, score of 0) versus completely intact (score of 10). We extract the forest integrity values using ArcGIS and calculate the individual point forest integrity value for each camera[21,60]. To create equal samples of comparatively intact versus comparatively disturbed forests for our kernel density estimations, we categorize camera locations based on median forest disturbance for all cameras at all landscapes (FLII$_{median}$ = 8.77; scatterplot shown in Supplementary Fig. 1). The binary split of all 1218 cameras based on median forest disturbance may limit the magnitude of behavioural shifts since disturbed habitat cameras with forest integrity 8.76 are differentiated from intact cameras with only slightly higher forest integrity values of 8.78. To investigate the sensitivity of our results to this split, we repeatall analyses after redefining split based onthe first quartile (FLII$_{Q1}$ = 5.61) and third quartile (FLII$_{Q3}$ = 9.73;

i.e., forest integrity ≤Q1 = "very disturbed"; forest integrity ≥Q3 = "very intact"). This approach halves the cameras available for analysis. There are unique disturbance histories for each landscape. Variation in the onset and duration of disturbances has been shown to strongly affect species richness (e.g., extinction debt in fragmented habitats) and may also impact species behaviour[61]. FLII is the most comprehensive proxy available yet the full effect of variable disturbance histories and hunting among our sites may not be captured[62].

## Assigning guilds

We assign the guild of each species based on their diet (carnivore, herbivore, or omnivore) and body mass using PanTHERIA for mammals, AVONET databases for birds[63,64] and published articles for reptiles[65–67] (species' trait data presented in Supplementary Table 2). We group body sizes as being 'small' (<4 kg; $N = 25$), 'medium' (4–20 kg; $N = 26$), and 'large' (>20 kg; $N = 12$) leading to groupings of two large carnivore species, six large herbivores, four large omnivores, eight medium carnivores, nine medium herbivores, nine medium omnivores, 12 small carnivores, seven small herbivores, and six small omnivores.

## Taxonomic inclusion criteria

When assessing changes in community-level diel activity, we include all vertebrates >1 kg detected from our cameras (Supplementary Table 2). We use genus-level identifications for three allopatric but ecologically similar congeners, clouded leopards (*Neofelis diardi* and *N. nebulosa* → *Neofelis spp.*), leopard cats (*Prionailurus bengalensis* and *P. javanensis* → *Prionailurus spp.*) and fireback pheasants (*Lophura rufa* and *L. ignita* → *Lophura spp.*)] as well as two mousedeer that are sympatric and ecologically similar congeners that cannot confidently be separated in many images (*Tragulus kanchil* and *T. napu* → *Tragulus spp.*)[68]. We repeat the analyses excluding the four most commonly abundant edge species to determine if any shifts are primarily due to species replacements (e.g., diurnal herbivore specialists with nocturnal omnivore generalists) or behavioural adaptations, the latter meaning the same species changing their activity. Over 53% of our total detections consist of two pig species, the wild boar and bearded pig (*Sus scrofa* and *S. barbatus*) and two species of macaques, the pig-tailed and long-tailed macaques (*Macaca nemestrina* and *M. fascicularis*). We note that long-tailed macaques are not common in intact forests but are frequently detected in disturbed forests (Supplementary Fig. 7; Supplementary Table 2). To assess species-level changes in diel activity using multinomial logistic regressions, we only select species with ≥15 detections in each diel category (i.e., day, twilight, and night; Supplementary Table 3). For the species-level analyses used to infer overlap among predator-prey and competitor pairs using kernel density estimation, we exclude species with <20 detections in both intact and disturbed forests (Supplementary Table 4). In Fig. 3, the "Change % intact → disturbed" is calculated within each diel category as: [(diurnal detections in disturbed habitats/total detections in disturbed habitats) − (diurnal detections in intact habitats/total detections in intact habitats)] *100.

## Computing activity distributions and overlap using kernel density estimation

We calculate activity distributions with circular kernel probability functions and 95% confidence intervals (CIs) with the fitact() function in the 'activity' package version 1.3.3 (bootstrapping 10,000 iterations)[69]. We calculate the coefficient of overlap (Δ) between activity distributions using the OverlapEST() function in the 'overlap' package version 0.3.4[59]. Δ denotes the shared area under the two activity distributions and ranges from 0 (no overlap) to 1 (full overlap). Following Ridout and Linkie[26], we report the $\Delta_4$ overlap estimator with a smoothing parameter of 1 when the sample size is >75 and the $\Delta_1$ overlap estimator with a smoothing parameter of 0.8

when the sample size is ≤75. We use the compareCkern() function found in the 'activity' package to assess the statistical significance between activity distributions[69]. This function is a randomization test that generates a null distribution of overlap indices using data sampled randomly with replacement from the combined datasets[26]. We also determine the activity peaks (AP) within each diel categories [i.e., day (0730–1630 h; 9 h total), twilight (0430–0730 h or 1630–1930 h; 6 h total) and night (1930–0430 h; 9 h total)] for each species, guild, and community, using the highest densities for each activity distribution. We then evaluate biologically meaningful behavioural shifts using two criteria: (1) a change in activity peak between day, twilight, or night, and (2) a p-value of <0.05 from the circular distribution randomisation test comparing the activity distributions in intact and disturbed forests. These two criteria account for either statistically significant and/or large shifts in activity peak within a single period (e.g., from 0900 h to 1530 h is a large shift within the same diurnal period) or small activity shifts that cross threshold among day, twilight and night diel categories (e.g., a change in peak activity from 0735 h to 0725 h).

## Multinomial logit mixed models to assess changes in diel activity

We use multinomial logit mixed models (MNLMMs) with three response variable categories (i.e., day, twilight, and night) to assess if the probability of wildlife detections occurring in each diel category changes in response to disturbance. We fit community-, guild-, and species-level models with forest integrity as our disturbance covariate. For our community-level models, we also add three other covariates of interests, body size (i.e., large, medium, and small), body mass (in kg) and feeding guild (i.e., carnivore, herbivore, and omnivore) as well as their interactions with forest integrity. For each community-, guild-, and species-level models, we calculate the Akaike Information Criterion (AIC) score to select the best model (i.e., model with the lowest AIC score) for each animal grouping. We include the observations landscape as a random effect in all models and used restricted maximum likelihood (REML). For the community- and guild-level models, we include species-level random effects in the main text results, where species contributions to the overall results are weighted similarly regardless of differences in the number of detections. To test how species turnover affected community- and guild-level results, we remove the species random effect thereby allowing detections to be weighted equally regardless of species which is shown in the supplementary materials. We set twilight as the reference category for all our models. We implement all MNLMMs in the 'mclogit' package in R[70] and plot the predicted probabilities for each diel category using the package "stats" version 4.3.1[71].

## Community and guild-level temporal interactions

To understand the change in species overlap within disturbed forests, we first extract activity distributions and Δ overlap for all competitor pairs and pairs of predators and their potential prey. We define species pairs as having potential 'competitive interactions' when species exhibit overlapping ranges and share the same body size and feeding category. Exceptions include all sizes of porcupines (both small and medium-sized), which are allowed to compete, and all sizes of strictly herbivorous ungulates [i.e., Malay tapir (*Tapirus Indicus*), sambar deer (*Rusa unicolor*), red muntjac (*Muntiacus muntjak*), and mouse deer], which are also allowed to compete. For apex predators, we allow tiger, leopard, and clouded leopards to compete because they are known to share prey species[42]. Predator-prey pairs are established for species with (i) overlapping ranges, (ii) at least one or both species possessing a predominantly carnivorous diet, and (iii) predators can only predate species of the same size category or lower (with an exception for medium-sized clouded leopards that are allowed to predate large ungulates, as noted above)[42]. We also assume that omnivores do not predate other species, since most omnivorous species within our

community primarily consume insects, fungi, carrion, and potentially animals <1 kg such as rodents and birds that are excluded from this study. We then calculate the species-level pairwise overlap in disturbed and intact forests and use a paired t-test to determine significant differences. Lastly, we use the Mann–Whitney test when species pairs are further split into their respective guilds and similarly determine significant differences between forest types. We note that no further statistical adjustments are carried out for all our analyses. We conduct all our analyses using the R statistical software version 4.2.0[71].

**Reporting summary**

Further information on research design is available in the Nature Portfolio Reporting Summary linked to this article.

## Data availability

The full camera trapping and species trait data collected for this study can be accessed via Figshare using the following DOI:[72] https://doi.org/10.6084/m9.figshare.23513412.

## code availability

The code used to model and visualise wildlife activity patterns can be obtained via GitHub using this link: https://github.com/EcologicalCascadesLab/WildlifeActivityPatterns

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

## Acknowledgements

We thank Yayasan Sabah, the Sabah Forest Department, the Sabah Biodiversity Council, the Danum Valley Management Committee, Glen Reynolds, and Jedediah Brodie for permission and help to conduct fieldwork at Danum Valley. We thank the Smithsonian Institute's Tropical Ecology Assessment and Monitoring (TEAM) network for help collecting data from Pasoh, as well as the Forest Research Institute Malaysia (FRIM) for permission to work there. We thank the Sarawak Forestry Department for permission to conduct fieldwork at Lambir Hills and Stuart Davies and the Nanyang Technological University Singapore field ecology courses for fieldwork help in Malaysia. We thank NParks for permission and help with fieldwork in Singapore. We thank Sarayudh Bunyavejchewin and the Thai Department of National Parks, Wildlife and Plant Conservation for permissions and help at Khao Yai and Khao Ban Tat. We thank Wido Rizqi Albert, Matthew Linkie, Yoan Dinata, Hariyo Wibisono and HarimauKita for help facilitating fieldwork in Sumatra, and we thank the Leuser International Foundation and WCS-Indonesia for assistance with fieldwork. Original artwork was provided courtesy of T. Barber from Talking Animals. We thank the members of the Ecological Cascades Lab at the University of Queensland, especially Calebe Mendes and Tom Bruce for comments that improved previous drafts. The research was funded by the Smithsonian Institution's ForestGEO program, Nanyang Technological University in Singapore, the University of Queensland (UQ), and National Geographic Society's Committee for the Research and Exploration award #9384–13. Kind support was provided by Fauna and Flora International (FFI), the Leuser International Foundation (LIF), Wildlife Conservation Society (WCS), and Southeast Asia Rainforest Research Partnership (SEARRP). MSL was supported by an Australian Research Council Discovery Early Career Researcher Award DECRA #DE210101440.

## Author contributions

S.L. proposed and conceptualized the hypotheses of the study presented here. S.L., M.S.L., and Z.A. developed the analytical approach of the study. S.L. developed and performed the analyses required for the study. M.S.L. approved and verified the analyses that has been

undertaken for this study. S.L., with the support of M.S.L., created and design the visualisations used in this study. S.L. wrote the manuscript with guidance and support from Z.A., K.M.G., and M.S.L. Camera trap data used in this study was deployed and collected by J.H.M. and M.S.L. M.S.L. provided the funding necessary for the implementation of this study.

## Competing interests

The authors declare no competing interests.
