## [Peer Review File · Nature Communications]

Effects of human disturbances on wildlife behaviour and consequences for predator-prey overlap in Southeast AsiaReviewer #1 (Remarks to the Author):

The authors conducted a large camera trapping study to examine the impacts of human activities on animal activity across Southeast Asia. They found shifts in peak activity patterns in response to humans in disturbed areas compared with undisturbed areas. These shifts differed depending on the diet of a species and how specialised a species is as a result of morphological and physiological differences. Interestingly, the authors found no change in the overlap between predators and prey, or competitor guilds. I commend the authors on their study. It provides a unique opportunity to examine behavioural shifts across different study areas and delve into the mechanisms underlying the patterns found by Gaynor et al. (2018) in *Science*. This study also complements a recent publication by Semper-Pascual et al. (2023) focusing on human impacts on tropical forest mammal occupancy dynamics using camera trapping data and also found differences between specialists and generalists (<https://doi.org/10.1038/s41559-023-02060-6>).

I had a couple of comments below.

In the abstract, I would suggest mentioning the general region of the study to indicate the applicability domain of the study to provide the readers with this context from the beginning.

Were the disturbed sites disturbed for similar periods of time? Is there a chance that some areas have been disturbed for longer periods and animals might have adapted or completely avoided these areas? This is touched on in lines 276 – 288, but I was wondering if this varies between sites in the study?

One small suggestion for FigS3: would it be possible to use the same configuration here as in Fig. 4 of the main text? This would make it easier to make direct comparisons between the different FLII treatments.

Reviewer #2 (Remarks to the Author):

This is an excellent paper on a topic that will be of interest to a broad international readership. The methods are sound and well described, the results are clear, and the conclusions well supported by the data.

I suggest some minor edits to improve clarity, as shown using track changes in the attached document. I hope these suggestions are helpful.

My only other comment concerns Fig. 3: unless I'm misreading the figure, I think the plus and minus signs are incorrect in panel c - shouldn't it be -6.4% and +9.0%?

Reviewer #3 (Remarks to the Author):

[Please see attached]

Review for:

Opposing effects of human disturbances on wildlife behavior and the consequences for predator-prey overlap

NOTE: I have written this review markdown format. I've also included a pdf if that is easier to read.

In this paper the authors used camera trap data to evaluate how human disturbance may be associated to shifts in predator-prey temporal overlap. They collected an impressive amount of data for a lot of species, and the paper is well written. However, I do have some concerns with the analysis and I am left a little unsure if their current analysis can be used to evaluate their hypotheses. My main concerns with the current manuscript were:

1. Are camera traps a robust method to capture diel activity patterns of birds? Certainly, there are some birds that are mostly flightless and large that camera traps would work great for (e.g., in North America the turkey comes to mind). However, other species may not spend their entire day on the ground (or towards the base of a tree), which limits how useful this method may be for birds (and even arboreal mammals).
2. Across all species there was a 3.2% change in activity patterns in intact vs disturbed areas. This resulted in a decrease in nocturnality in disturbed areas and therefore increases in day time (1.2%) and twilight (2.0%) activity. Given sample sizes this was pretty easily identified as a significant difference, but I'm left wondering whether a 3% shift is biologically significant.
3. I have some concerns with respect to the current community level estimation of activity patterns. Even with the removal of some common species like pigs and macaques, the authors make the assumption that all diel activity records are i.i.d. and are drawn from the same von mises distribution. Yet, as the authors can attest to, species not only vary in their diel activity (less of a concern here with respect to estimation), but also do so in response to the other species in their environment. On top of this, sample sizes vary among species, and as a result some species will have more weight with respect to the overall pattern (and removing pigs and macaques does not address this for the rest of the community). This makes the overall estimation of community level patterns a little suspect and very difficult to interpret. Some sort of hierarchical parameterization for circular kernel density estimates would address this, but to my knowledge such a model has not been something that ecologists have really used yet (though other fields have used them, but see the following comment with more specific modeling suggestions).
4. As it appears that the predominant interest here is variation in selection for different diel categories (i.e., day, twilight, and night), then a natural modeling choice to account for variation among species (or guilds) would be multinomial model where your response variable is the number of detections in each diel category by species. Modeling in this way would allow the authors to account for variation among species using hierarchical structure (i.e., random effects), variation among sampling areas (the 10 forests), but also include FLII as a continuous measure instead of using an arbitrary cutoff of 8.77 to classify a forest as intact or disturbed. Certainly, kernel density estimates could still be used to visualize the patterns, but if you want to know if

nocturnality increased or decreased, a multinomial regression is an ideal choice (see Gallo et al. 2022). Essentially, while the kernel density estimates are easy to calculate, they make it impossible to account for some aspects of the data as they have been presented, which makes it difficult to determine whether or not the authors can evaluate their hypotheses with the current analysis..

Gallo, T., Fidino, M., Gerber, B., Ahlers, A. A., Angstmann, J. L., Amaya, M., ... & Magle, S. B. (2022). Mammals adjust diel activity across gradients of urbanization. *Elife*, 11, e74756.

5. Interactions occur in both space and time. To me, however, it seems as if the authors only evaluated 'overlap' over time, and assumed species occupied the entire space. In other words, the authors generated activity estimates for predators & prey across all camera trapping sites and then determined their overlap (based on my reading of the methods). This does not seem like the most appropriate evaluation of the authors hypothesis given that it does not account for predator presence in the prey activity estimate (for example). In other words, I think the authors should be comparing the prey species activity pattern given the presence / absence of a given predator (e.g., what is the overlap for a subordinate species at sites that do and do not have the dominant species?). Currently, not seeing any big shifts in overlap may just be because the species are spatially segregated (and as such there is no need for shifts in temporal activity).

In the section below I've included top-level thoughts for each section as well as line by line comments. I hope the authors find them useful. If the authors have any specific questions about anything I've put in this review I can be reached at mfidino@lpzoo.org

- Mason Fidino

Abstract

Top-level thoughts

1. The paper does not actually evaluate how morphological or physiological constraints drive diel behavior. The abstract does use the qualifier `may` (line 28) when talking about morphological / physiological constraints, but from my reading of the abstract I thought such traits were going to be evaluated in the paper itself. This could 100% be a 'me' thing, just something that caught me off guard after reading through the paper a second time.

Introduction

Top-level thoughts

1. Currently the authors assume that all readers know what diel activity is (line 41). I'm definitely in this camp, but to reach a more general audience it would help to add a very brief definition when it's introduced on line 41. Something like:

One example is how humans may impact wildlife diel activity, or how species use the 24 hour light dark cycle, which stems from historical and contemporary hunting that instilled a fear of humans as predators or aggressive competition in most animals.

2. The last paragraph ends in such a way that it seems like more information is going to be shared as the last sentence is a bit of info on how the authors quantified intact vs disturbed forests. Currently, the paragraph provides some methodological details but there is no distillation of that information to the reader at the end of the paragraph.

Line by line comments

Line 39: Could remove the gerund here and just say "..., which impacts wildlife community..."

Line 40: unclear who is doing the impacts here. Do you mean anthropogenic impacts or impacts in general? Could help a bit to be explicit here and say anthropogenic impacts .

Line 41: historic generally means an event that was important to history, historical means something that happened in the past. I'm guessing you mean historical here.

Line 53: It would help to be explicit about why this a biologically meaningful difference. Right now the reader has to take the authors word for it, which is not ideal.

Line 64: I think you mean increased temporal overlap instead.

Line 70: drop the parentheses in the aside split apart with em dashes.

Line 71 - 73: I'm not sure the point the authors are trying to get across here is. Perhaps it is because 'potential likelihood of interactions' does not really have a solid definition here. On top of this, the authors have not shared any logic here about why this limited interpretation is useful for understanding community dynamics. Some more logic here would help.

Line 75: to evaluate instead of for evaluating .

Line 78: edges of what?

Results

Top-level thoughts

1. I tried my best to link the results with each hypothesis, and unfortunately it seems like many of the hypotheses are not explicitly evaluated (save for H4).

(H1) species that are diurnal and crepuscular in intact habitats significantly shift their behaviour towards nocturnality in disturbed habitats to avoid diurnal humans,

I do not see any results that specifically evaluate this. For example, there is no framework to classify species as diurnal or crepuscular in intact habitats, and as such it is unclear whether those species became more nocturnal. Even Fig. S5 can't

really demonstrate this given that the hypothesis is conditional on diel behavior in intact habitats.

(H2) differences in the community-and guild-level activity pattern are driven by winner-loser species replacements (e.g., intact habitat specialists that are diurnal being replaced by nocturnal generalists in disturbed habitats),

There is no evaluation about the species who make up community and guild-level activity patterns, though there are guild-level results starting on line 166. The authors state that the macaque and pig results provide evidence for this hypothesis (see lines 239 - 241 in the discussion), but there was no evaluation of replacement in the analysis (i.e., some species were filtered out and others have taken their place).

(H3) hunters induce more fear in preferred species (larger game animals) driving stronger shifts compared to smaller and non-target species

To me, I'm not seeing an explicit connection between this hypotheses and the reported results. For example, the authors presented guild-level results about animals that vary in body size, but there is no direct statistical evaluation of this hypothesis, which to me would require evaluating shifts in diel activity as a function of things like body size (which again could be done with a multinomial model, but not really with circular kernel density estimates).

(H4) humans repel wildlife from diurnal hours driving increase temporal overlap during nocturnal periods
among predator-prey and competitor species pairs

Assuming that disturbed forests have more humans, then there is some evaluation about competition and predation (see Figure 5) as well as in Figure S5.

As a result, the paper is not near as cohesive as it could be, as the results could be better framed in the context of the hypotheses the authors shared (though perhaps some different analyses must be done to more explicitly evaluate these hypotheses). For example, if there are four hypotheses, then having sub-sections in the results that are explicitly about each hypothesis would help.

Line by line comments

Line 111-112: seems like this sentence is missing a word.

Line 171: there is a comma after an em dash.

Line 216 - 231: Great job adding this sensitivity analysis! Splitting by the average always feels a little arbitrary, so it is refreshing to see the authors consider how this splitting procedure may have an influence on some of their results.

Discussion

Top-level thoughts

1. How did the authors evaluate statistical differences of different diel phenotypes (diurnal to nocturnal for a single species)? There is a lot of sentences of this in the first paragraph of the discussion but I saw none of that in the results.

2. With four hypotheses, it may help to have just separate paragraphs for each in the order you presented them in the introduction. The 2nd paragraph jumps right to H3 and I don't really think that H1 and H2 have been fully unpacked in the first paragraph.

Line by line comments

Line 235 - 237: predict interactions and composition of what?

Line 241 - 245: This all seems like it should be in the results.

Line 251 - 252: This was not really evaluated in your analysis.

Line 251 - 260: The discussion here demonstrates some of the weaknesses associated to kernel density estimates. As they are just descriptions of the data, it is difficult for the reader to really understand what the authors mean with statements like "Namely, mid-sized carnivores deviated from the community pattern by markedly increasing their diurnal activity." (lines 256 - 259). Certainly, the kernel density estimates visually look different, but there is no quantification of these differences (e.g., effect size of the differences), nor is there uncertainty estimates provided.

Methods

Top-level thoughts

1. It's actually less reproducible to share which R function you used rather than the specific analysis done (Edwards and Auger-Méthé 2019). For example, I may have a different version of `overlap` on my computer (v 0.3.4) than the authors and it does not contain the `fitact()` function. Similarly, I have an `overlapEst()` function, but not an `overlapEST()` function. It's a very real possibility that R may not be the programming language we all use 20 years from now, so instead it is more important to describe the mathematical methodology rather than the R function you used. Fortunately, you do a great job describing the statistical method in combination with the R functions, so really you just need to add which version of `overlap` you used and double check the function names to ensure they are correct (also there is a space right after `overlapEST` on line 366 that does not need to be there).

Edwards, A. M., & Auger-Méthé, M. (2019). Some guidance on using mathematical notation in ecology. *Methods in Ecology and Evolution*, 10(1), 92-99.

2. On assessing shifts in activity distributions. I'm not especially convinced that the described method the authors provide on lines 378 - 387 is appropriate as it does not explicitly assess changes in activity (e.g., diurnal to nocturnal). Essentially, there is no statistical test that occurs which explicitly evaluates whether a species has a change in activity patterns in disturbed forests conditional on their activity in intact forests. In other words, given that a species nocturnal what is the probability they shift their activity to the other diel periods? Instead, the authors have used their diel activity data twice, first to determine the activity peaks and then to determine if diel activity patterns differ between intact and disturbed forests for each species. The first test is a descriptive statistic of their data, while the second test does not evaluate a targeted shift, rather it evaluates

if diel patterns differ. As such, the authors cannot determine if it is this specific shift in peak activity that is the cause of this difference, which is what I think the authors want to do.

Line by line comments

Line 381: What is a biologically meaningful behavioral shift?

Line 384: You describe two criteria and then say three here.

Tables & figures

Top-level thoughts

Figure 1: It's a little odd that one human is a silhouette while all the other anthropogenic impacts are line drawings with different styles. I'm also not understanding figure a) here. Also, what do the arrows represent in all of the sub-figures? Also define what the sun, half sun, and moon shapes represent, I'm guessing dawn, twilight, and night? Finally, what is the difference between the green forest on the bottom of a (or top of b) and the black forest on the right of a (or bottom of b). In figure a it seems like a shift from day to night, while in figure b it seems like shift from intact to disturbed.

Figure 5. What are the larger dots and error bars? Furthermore, I think the coloring and ordering on 5d hides the specific point that authors are trying to get across: that within a species 'type of predation' there is no difference in overlap between intact and disturbed forests. The authors have colored, for example, carnivore-carnivore in two different colors and have separated the intact and disturbed estimates from one another. I think it would be more clear if the points were colored for 'intact' and 'disturbed' and then on the x axis you would have the predation types listed. That way the reader can more easily make the comparisons that the authors want to demonstrate.

Reviewer #1 (Remarks to the Author):

The authors conducted a large camera trapping study to examine the impacts of human activities on animal activity across Southeast Asia. They found shifts in peak activity patterns in response to humans in disturbed areas compared with undisturbed areas. These shifts differed depending on the diet of a species and how specialised a species is as a result of morphological and physiological differences. Interestingly, the authors found no change in the overlap between predators and prey, or competitor guilds. I commend the authors on their study. It provides a unique opportunity to examine behavioural shifts across different study areas and delve into the mechanisms underlying the patterns found by Gaynor et al. (2018) in *Science*. This study also compliments a recent publication by Semper-Pascual et al. (2023) focusing on human impacts on tropical forest mammal occupancy dynamics using camera trapping data and also found differences between specialists and generalists (<https://doi.org/10.1038/s41559-023-02060-6>).

I had a couple of comments below.

****Response:** Thank you, we agree and we have now edited the discussion to explicitly mention the Semper-Pascual paper.

In the abstract, I would suggest mentioning the general region of the study to indicate the applicability domain of the study to provide the readers with this context from the beginning.

****Response:** Thank you for pointing this out. We have specified it in Line 24 of the Abstract.

Were the disturbed sites disturbed for similar periods of time? Is there a chance that some areas have been disturbed for longer periods and animals might have adapted or completely avoid these areas? This is touched on in lines 276 – 288, but I was wondering if this varies between sites in the study?

****Response:** Good point. We have added the following sentence to the Methods section:

“There are unique disturbance histories for each landscape. Variation in the onset and duration of disturbances has been shown to strongly affect species richness (e.g., extinction debt in fragmented habitats) and may also impact species behaviour (Betts *et al.* 2019). While FLII is the most comprehensive proxy available, the full effect of variable disturbance histories among our sites may not be captured.”

Betts *et al.* Extinction filters mediate the global effects of habitat fragmentation on animals. *Science* **366**, 1236-1239 (2019).

One small suggestion for FigS3: would it be possible to use the same configuration here as in Fig. 4 of the main text? This would make it easier to make direct comparisons between the different FLII treatments.

****Response:** Great suggestion, we edited to be the same configuration now.

Reviewer #2 (Remarks to the Author):

This is an excellent paper on a topic that will be of interest to a broad international readership. The methods are sound and well described, the results are clear, and the conclusions well supported by the data.

I suggest some minor edits to improve clarity, as shown using track changes in the attached document. I hope these suggestions are helpful.

****Response:** Thank you for your suggestions. They have been helpful.

My only other comment concerns Fig. 3: unless I'm misreading the figure, I think the plus and minus signs are incorrect in panel c - shouldn't it be -6.4% and +9.0%?

****Response:** We double checked the percentages.

Reviewer #3 (Remarks to the Author):

In this paper the authors used camera trap data to evaluate how human disturbance may be associated to shifts in predator-prey temporal overlap. They collected an impressive amount of data for a lot of species, and the paper is well written. However, I do have some concerns with the analysis and I am left a little unsure if their current analysis can be used to evaluate their hypotheses. My main concerns with the current manuscript were:

1. Are camera traps a robust method to capture diel activity patterns of birds? Certainly, there are some birds that are mostly flightless and large that camera traps would work great for (e.g., in North America the turkey comes to mind). However, other species may not spend their entire day on the ground (or towards the base of a tree), which limits how useful this method may be for birds (and even arboreal mammals).

****Response:** The bird species included are terrestrial phasianidae family species that are consistently caught by our cameras. We only included species that has a body mass of more than 1 kg which filters off many of the smaller and more volant bird species (see Methods section 2.3). We address this in the manuscript.

2. Across all species there was a 3.2% change in activity patterns in intact vs disturbed areas. This resulted in a decrease in nocturnality in disturbed areas and therefore increases in day time (1.2%) and twilight (2.0%) activity. Given sample sizes this was pretty easily identified as a significant difference, but I'm left wondering whether a 3% shift is biologically significant.

****Response:** The new MNLMM approach provides better interpretations of biological significance. We have added a specific sentence about biological significance in the new MNLMM results section: "However, focusing on this level of statistical significance overlooks biologically important differences among the specialists versus generalist groupings. Namely, the rarer 59 specialists' probability of nocturnality in the most disturbed sites [Pr(night) at FLII of 0] was 4.6 times higher than for pigs and macaques [Pr(night)rarer_specialists = 0.41, CI = 0.39 – 0.43; Pr(night)pigs_macaques = 0.089, CI = 0.066 – 0.11], and specialists' nocturnality increased in the most intact forest (FLII of 10) to 6.4 times higher than pigs and macaques [Pr(night)rarer_specialists = 0.46, CI = 0.44 – 0.48; Pr(night)pigs_macaques = 0.072, CI = 0.052 – 0.092]."

3. I have some concerns with respect to the current community level estimation of activity patterns. Even with the removal of some common species like pigs and macaques, the authors make the assumption that all diel activity records are i.i.d. and are drawn from the same von mises distribution. Yet, as the authors can attest to, species not only vary in their diel activity (less of a concern here with respect to

estimation), but also do so in response to the other species in their environment. On top of this, sample sizes vary among species, and as a result some species will have more weight with respect to the overall pattern (and removing pigs and macaques does not address this for the rest of the community). This makes the overall estimation of community level patterns a little suspect and very difficult to interpret. Some sort of hierarchical parameterization for circular kernel density estimates would address this, but to my knowledge such a model has not been something that ecologists have really used yet (though other fields have used them, but see the following comment with more specific modeling suggestions).

****Response:** Thank you for your concern in regards to our current modelling approach. We have taken your advice and implemented a multinomial logit mixed modelling (MNLMM) approach to analyse the results. Please see the following responses for further details and revised manuscript.

4. As it appears that the predominant interest here is variation in selection for different diel categories (i.e., day, twilight, and night), then a natural modeling choice to account for variation among species (or guilds) would be multinomial model where your response variable is the number of detections in each diel category by species. Modeling in this way would allow the authors to account for variation among species using hierarchical structure (i.e., random effects), variation among sampling areas (the 10 forests), but also include FLII as a continuous measure instead of using an arbitrary cutoff of 8.77 to classify a forest as intact or disturbed. Certainly, kernel density estimates could still be used to visualize the patterns, but if you want to know if nocturnality increased or decreased, a multinomial regression is an ideal choice (see Gallo et al. 2022). Essentially, while the kernel density estimates are easy to calculate, they make it impossible to account for some aspects of the data as they have been presented, which makes it difficult to determine whether or not the authors can evaluate their hypotheses with the current analysis..

...

Gallo, T., Fidino, M., Gerber, B., Ahlers, A. A., Angstmann, J. L., Amaya, M., ... & Magle, S. B. (2022). Mammals adjust diel activity across gradients of urbanization. *Elife*, 11, e74756.

...

****Response:** Thank you for the insightful suggestion. We adopted this modelling approach and revised added this to the Methods section 2.6 *Multinomial models to assess changes in diel activity*. We have also revised our results for both the community- and guild-level diel shifts and replaced the main figures.

5. Interactions occur in both space and time. To me, however, it seems as if the authors only evaluated 'overlap' over time, and assumed species occupied the entire space. In other words, the authors generated activity estimates for predators & prey across all camera trapping sites and then determined their overlap (based on my reading of the methods). This does not seem like the most appropriate evaluation of the authors hypothesis given that it does not account for predator presence in the prey activity estimate (for example). In other words, I think the authors should be comparing the prey species activity pattern given the presence / absence of a given predator (e.g., what is the overlap for a subordinate species at sites that do and do not have the dominant species?). Currently, not seeing any big shifts in overlap may just be because the species are spatially segregated (and as such there is no need for shifts in temporal activity).

****Response:** Spatiotemporal overlap is certainly interesting and the next step in the evolution of statistics for quantifying species interactions from observation datasets such as camera traps. While our group is working on ways to conduct such analyses (e.g. see Amir et al 2023), there are no solutions yet available. We may reach out to this reviewer separately to discuss designing new methodological approaches to quantify species interactions that include both space and time.

Amir, Z., Sovie, A. and Luskin, M.S., 2022. Inferring predator–prey interactions from camera traps: A Bayesian co- abundance modeling approach. *Ecology and Evolution*, 12(12), p.e9627.

In the section below I've included top-level thoughts for each section as well as line by line comments. I hope the authors find them useful. If the authors have any specific questions about anything I've put in this review I can be reached at mfidino@lpzoo.org

- Mason Fidino

Abstract

Top-level thoughts

1. The paper does not actually evaluate how morphological or physiological constraints drive diel behavior. The abstract does use the qualifier `may` (line 28) when talking about morphological / physiological constraints, but from my reading of the abstract I thought such traits were going to be evaluated in the paper itself. This could 100% be a 'me' thing, just something that caught me off guard after reading through the paper a second time.

****Response:** Agreed, we removed this from the Abstract.

Introduction

Top-level thoughts

1. Currently the authors assume that all readers know what diel activity it (line 41). I'm definitely in this camp, but to reach a more general audience it would help to add a very brief definition when its introduced on line 41. Something like:

...

One example is how humans may impact wildlife diel activity, or how species use the 24 hour light dark cycle, which stems from historical and contemporary hunting that instilled a fear of humans as predators or aggressive competition in most animals.

...

****Response:** Thank you for your kind suggestion. Agreed, we have included a short definition as suggested.

2. The last paragraph ends in such a way that it seems like more information is going to be shared as the last sentence is a bit of info on how the authors quantified intact vs disturbed forests. Currently, the paragraph provides some methodological details but there is no distillation of that information to the reader at the end of the paragraph.

****Response:** Nature Communications has a unique style where the editors request a summary of the main results at the end of the Introduction. Accordingly, we added two summary sentences and that also addresses this specific comment by the reviewer.

Line by line comments

Line 39: Could remove the gerund here and just say "..., which impacts wildlife community..."

****Response:** Thank you for the suggestion. We have edited it accordingly.

Line 40: unclear who is doing the impacts here. Do you mean anthropogenic impacts or impacts in general? Could help a bit to be explicit here and say `anthropogenic impacts`.

****Response:** Thank you for pointing this out. We have specified it and now it reads as "anthropogenic impacts".

Line 41: historic generally means an event that was important to history, historical means something that happened in the past. I'm guessing you mean historical here.

****Response:** Thank you for pointing this out. Yes, we did mean to write "historical" here and have changed it accordingly in the manuscript.

Line 53: It would help to be explicit about why this a biologically meaningful difference. Right now the reader has to take the authors word for it, which is not ideal.

****Response:** Thank you for pointing this out. We have now included a deeper discussion of why small changes in diel activity can have biologically meaningful ramifications for species fitness in Line 57.

Line 64: I think you mean `increased` temporal overlap instead.

****Response:** Yes, this was edited accordingly.

Line 70: drop the parentheses in the aside split apart with em dashes.

****Response:** We have changed it accordingly in the manuscript. Thank you.

Line 71 - 73: I'm not sure the point the authors are trying to get across here is. Perhaps it is because 'potential likelihood of interactions' does not really have a solid definition here. On top of this, the authors have not shared any logic here about why this limited interpretation is useful for understanding community dynamics. Some more logic here would help.

****Response:** We believe that readers will intuit that species overlap - being active at the time same - increases the likelihood of interactions. We provided the logic in the preceding paragraph line 55: "This distinction is biologically meaningful, as crepuscular and nocturnal periods offer different light and thermal environments¹⁴. Specifically, the semi-dark environment during crepuscular periods possesses moderate light and thermal conditions, potentially balancing the benefits of visibility for foraging and thermal stress associated with thermoregulation^{14,15}, but species can still be spotted by diurnal predators with vision are adapted to light. Conversely, the darkness of nocturnal periods can provide a natural protection from light-

adapted predators, however, species can incur fitness costs from thermoregulation and reduced foraging efficiency and difficulty spotting nocturnal predators due to lower visibility^{14, 15}. Therefore, behavioural adaptations to avoid diurnal human can influence a species' ability to forage, evade predators, thermoregulate and ultimately survive."

Line 75: `to evaluate` instead of `for evaluating`.

****Response:** Thank you, changed.

Line 78: edges of what?

****Response:** Edited to be "forest edges".

Results

Top-level thoughts

1. I tried my best to link the results with each hypothesis, and unfortunately it seems like many of the hypotheses are not explicitly evaluated (save for H4).

...

(H1) species that are diurnal and crepuscular in intact habitats significantly shift their behaviour towards nocturnality in disturbed habitats to avoid diurnal humans,

...

I do not see any results that specifically evaluate this. For example, there is no framework to classify species as diurnal or crepuscular in intact habitats, and as such it is unclear whether those species became more nocturnal. Even Fig. S5 can't really demonstrate this given that the hypothesis is conditional on diel behavior in intact habitats.

****Response:** We have taken your advice and implemented multinomial models to assess changes in diel activity at all levels.

...

(H2) differences in the community-and guild-level activity pattern are driven by winner-loser species replacements (e.g., intact habitat specialists that are diurnal being replaced by nocturnal generalists in disturbed habitats),

...

There is no evaluation about the species who make up community and guild-level activity patterns, though there are guild-level results starting on line 166. The authors state that the macaque and pig results provide evidence for this hypothesis (see lines 239 - 241 in the discussion), but there was no evaluation of replacement in the analysis (i.e., some species were filtered out and others have taken their place).

****Response:** We have now addressed this specifically using the multinomial approach suggested by the reviewer.

...

(H3) hunters induce more fear in preferred species (larger game animals) driving stronger shifts compared to smaller and non-target species

...

To me, I'm not seeing an explicit connection between this hypotheses and the reported results. For example, the authors presented guild-level results about animals that vary in body size, but there is no direct statistical evaluation of this hypothesis, which to me would require evaluating shifts in diel activity as a function of things like body size (which again could be done with a multinomial model, but not really with circular kernel density estimates).

****Response:** We have now addressed this specifically in the manuscript using the multinomial approach suggested by the reviewer.

...

(H4) humans repel wildlife from diurnal hours driving increase temporal overlap during nocturnal periods among predator-prey and competitor species pairs

...

Assuming that disturbed forests have more humans, then there is some evaluation about competition and predation (see Figure 5) as well as in Figure S5.

As a result, the paper is not near as cohesive as it could be, as the results could be better framed in the context of the hypotheses the authors shared (though perhaps some different analyses must be done to more explicitly evaluate these hypotheses). For example, if there are four hypotheses, then having subsections in the results that are explicitly about each hypothesis would help.

****Response:** We have reformatted the Results and Discussion to be more aligned with the hypotheses.

Line by line comments

Line 111-112: seems like this sentence is missing a word.

****Response:** Thank you for pointing this out. We have included the missing word "were" in Line 112.

Line 171: there is a comma after an em dash.

****Response:** We have included the comma. Thank you for pointing this out.

Line 216 - 231: Great job adding this sensitivity analysis! Splitting by the average always feels a little arbitrary, so it is refreshing to see the authors consider how this splitting procedure may have an influence on some of their results.

****Response:** Thank you for the appraisal of our results.

Discussion

Top-level thoughts

1. How did the authors evaluate statistical differences of different diel phenotypes (diurnal to nocturnal for a single species)? There is a lot of sentences of this in the first paragraph of the discussion but I saw none of that in the results.

****Response:** We have now addressed this specifically using the multinomial approach suggested by the reviewer.

2. With four hypotheses, it may help to have just separate paragraphs for each in the order you presented them in the introduction. The 2nd paragraph jumps right to H3 and I don't really think that H1 and H2 have been fully unpacked in the first paragraph.

****Response:** Good suggestions. We labeled results to the four hypotheses (H1-H4) directly in the first Discussion paragraph now. Very clear.

Line by line comments

Line 235 - 237: predict interactions and composition of what?

****Response:** Thank you for pointing this out. We have change the sentence to read as: "...predict interactions and composition of species."

Line 241 - 245: This all seems like it should be in the results.

****Response:** We have moved them to the results.

Line 251 - 252: This was not really evaluated in your analysis.

****Response:** removed.

Line 251 - 260: The discussion here demonstrates some of the weaknesses associated to kernel density estimates. As they are just descriptions of the data, it is difficult for the reader to really understand what the authors mean with statements like "Namely, mid-sized carnivores deviated from the community pattern by markedly increasing their diurnal activity." (lines 256 - 259). Certainly, the kernel density estimates visually look different, but there is no quantification of these differences (e.g., effect size of the differences), nor is there uncertainty estimates provided.

****Response:** We have now addressed this statistically using the multinomial approach suggested by the reviewer.

Methods

Top-level thoughts

1. It's actually less reproducible to share which R function you used rather than the specific analysis done (Edwards and Auger- Méthé 2019). For example, I may have a different version of `overlap` on my computer (v 0.3.4) than the authors and it does not contain the `fitact()` function. Similarly, I have an `overlapEst()` function, but not an `overlapEST()` function. It's a very real possibility that R may not be the programming language we all use 20 years from now, so instead it is more important to describe the mathematical methodology rather than the `R` function you used. Fortunately, you do a great job describing the statistical method in combination with the R functions, so really you just need to add which version of `overlap` you used and double check the function names to ensure they are correct (also there is a space right after overlapEST on line 366 that does not need to be there).

...

Edwards, A. M., & Auger- Méthé, M. (2019). Some guidance on using mathematical notation in ecology. *Methods in Ecology and Evolution*, 10(1), 92-99.

...

****Response:** Thank you for pointing this out. We have included the version of each package accordingly. We have also remove the space right after overlapEST on line 366.

2. On assessing shifts in activity distributions. I'm not especially convinced that the described method the authors provide on lines 378 - 387 is appropriate as it does not explicitly assess changes in activity (e.g., diurnal to nocturnal). Essentially, there is no statistical test that occurs which explicitly evaluates whether a species has a change in activity patterns in disturbed forests conditional on their activity in intact forests. In other words, given that a species nocturnal what is the probability they shift their activity to the other diel periods? Instead, the authors have used their diel activity data twice, first to determine the activity peaks and then to determine if diel activity patterns differ between intact and disturbed forests for each species. The first test is a descriptive statistic of their data, while the second test does not evaluate a targeted shift, rather it evaluates if diel patterns differ. As such, the authors cannot determine if it is this specific shift in peak activity that is the cause of this difference, which is what I think the authors want to do.

****Response:** As mentioned above, we have now taken your advice and implemented a multinomial modelling approach. Please review Lines 487 - 499 of the manuscript for the revised methods of analysing changes in diel activity.

Line by line comments

Line 381: What is a biologically meaningful behavioral shift?

****Response:** this is ambiguous so we removed it. However, it was previously defined in the Methods as "We then evaluated biologically meaningful behavioural shifts] using two criteria: (1) a change in activity peak between day, twilight, or night, and (2) a p-value of < 0.05 from the circular distribution randomisation test comparing the activity distributions in intact and disturbed forests."

Line 384: You describe two criteria and then say three here.

****Response:** Sorry, we have changed it to "two" in Line 384.

Tables & figures

Top-level thoughts

Figure 1: It's a little odd that one human is a silhouette while all the other anthropogenic impacts are line drawings with different styles. I'm also not understanding figure `a)` here. Also, what do the arrows represent in all of the sub-figures? Also define what the sun, half sun, and moon shapes represent, I'm guessing dawn, twilight, and night? Finally, what is the difference between the green forest on the bottom of a (or top of b) and the black forest on the right of a (or bottom of b). In figure a it seems like a shift from day to night, while in figure b it seems like shift from intact to disturbed.

****Response:** Thank for your kind suggestions. We have included a line drawing of a hunter to match the other line drawings. We have also removed the forests below Figure 1a to avoid any confusions. We have added a revised figure caption which hopefully addresses your comments above. It reads, "**Figure 1.** Hypothesized wildlife behavioural response to humans and implications for species interactions. Grey animals show species that are present in intact forests but disappear or shift their activity in degraded forest. (a) Prior work suggests diurnal and crepuscular species alter their behaviour towards nocturnality in disturbed forests to avoid diurnal humans. The "sun", "half-sun" and "moon" symbols represent day, twilight and night respectively. (b) Community- and guild-level temporal shifts may be driven by winner-loser species replacements within disturbed forests (e.g., loss of diurnal specialists or defaunation of game species alongside an increase in omnivorous generalists). The panel shows four habitat specialists in intact forests ("green" forest) being replaced by four generalists (pigs and macaques) with disturbed forests ("red" forest). (c) Larger hunted animals may show stronger avoidance of humans. (d, e) Increased nocturnality may increase the temporal overlap of predator-prey and competitive species pairs. All arrows in (a), (c), (d) and (e) refer to the change in diel activity from day to night as forests become more disturbed while the arrow in (b) refers to the replacement of specialist by generalist species within disturbed forests."

Figure 5. What are the larger dots and error bars? Furthermore, I think the coloring and ordering on 5d hides the specific point that authors are trying to get across: that within a species 'type of predation' there is no difference in overlap between intact and disturbed forests. The authors have colored, for example, carnivore-carnivore in two different colors and have separated the intact and disturbed estimates from one another. I think it would be more clear if the points were colored for 'intact' and 'disturbed' and then on the x axis you would have the predation types listed. That way the reader can more easily make the comparisons that the authors want to demonstrate.

****Response:** Agreed, we made the suggested changes, thank you!

Reviewer #3 (Remarks to the Author):

The authors did a wonderful job on the revised manuscript. The paper was already great, but the additional analyses and organization of the results makes everything much more clear. I only have two small things that I feel should be looked at.

First, Figure 4 (the multinomial model results) has some sub-plots that look a little off. The separate probabilities should sum to 1 at each point along the x axis, and 4(d) looks like this is not the case, especially as some of the diel categories run off the y axis. I would just double check how you are making the predictions here to ensure that they are displayed correctly. Most of these models use the softmax function, which is just a generalization of the logit link for n categories.

Second, for the multinomial logit models the results discuss numerous statistical interaction terms being present in the top model. However, there was no reference to how models were constructed in the methods. Instead, the authors said they used AIC to assess the support of a few different covariates. I would encourage the authors to use a couple sentences to more fully flush out how they constructed their models.

Again, this paper was a pleasure to read, you should be proud of this quality piece of research!

Cheers,
Mason Fidino

Response to Reviewer 3

Reviewer #3 (Remarks to the Author):

The authors did a wonderful job on the revised manuscript. The paper was already great, but the additional analyses and organization of the results makes everything much clearer. I only have two small things that I feel should be looked at.

First, Figure 4 (the multinomial model results) has some sub-plots that look a little off. The separate probabilities should sum to 1 at each point along the x axis, and 4(d) looks like this is not the case, especially as some of the diel categories run off the y axis. I would just double check how you are making the predictions here to ensure that they are displayed correctly. Most of these models use the softmax function, which is just a generalization of the logit link for n categories.

****Response:** Thank you for pointing this out for us. We have re-calculated and made the necessary corrections. The probabilities of the different categories should add up to 1 now for Figure 4d.

Second, for the multinomial logit models the results discuss numerous statistical interaction terms being present in the top model. However, there was no reference to how models were constructed in the methods. Instead, the authors said they used AIC to assess the support of a few different covariates. I would encourage the authors to use a couple sentences to more fully flush out how they constructed their models.

****Response:** Thank you for your recommendation. We have since added more keywords to describe the model building process. Now, it reads, "We use multinomial logit mixed models (MNLMMs) with three response variable categories (i.e., day, twilight, and night) to assess if the probability of wildlife detections occurring in each diel category changes in response to disturbance. We fit community-, guild-, and species-level models with forest integrity as our disturbance covariate. For our community-level models, we also add three other covariates of interests, body size (i.e., large, medium, and small), body mass (in kg) and feeding guild (i.e., carnivore, herbivore, and omnivore) as well as their interactions with forest integrity. For each community-, guild-, and species-level models, we calculate the Akaike Information Criterion (AIC) score to select the best model (i.e., model with the lowest AIC score) for each animal grouping. We include the observations landscape as a random effect in all models and used restricted maximum likelihood (REML). For the community- and guild-level models, we include species-level random effects in the main text results, where species contributions to the overall results are weighted similarly regardless of differences in the number of detections. To test how species turnover affected community- and guild-level results, we remove the species random effect thereby allowing detections to be weighted equally regardless of species which is shown in the supplementary materials. We set twilight as the reference category for all our models. We implement all MNLMMs in the 'mclgfit' package in R and plot the predicted probabilities for each diel category using the package "stats" version 4.3.1".

Again, this paper was a pleasure to read, you should be proud of this quality piece of research!